

# Data Generated During the 2018 LAPSE-RATE Campaign: An Introduction and Overview

Gijs de Boer[1,2], Adam Houston[3], Jamey Jacob[4], Phillip B. Chilson[5], Suzanne W. Smith[6], Brian
Argrow[7], Dale Lawrence[7], Jack Elston[8], David Brus[9], Osku Kemppinen[10], Petra Klein[5,11], Julie K.
Lundquist[12,13], Sean Waugh[11], Sean C.C. Bailey[6], Amy Frazier[14], Michael P. Sama[6], Christopher
Crick[5], David Schmale III[15], James Pinto[16], Elizabeth A. Pillar-Little[5], Victoria Natalie[4], Anders
Jensen[16]

[1] Cooperative Institute for Research in Environmental Sciences, University of Colorado Boulder, Boulder, Colorado, USA
[2] NOAA Physical Sciences Laboratory, Boulder Colorado, USA
[3] University of Nebraska, Lincoln, Nebraska, USA
[4] Oklahoma State University, Stillwater, Oklahoma, USA
[5] University of Oklahoma, Norman, Oklahoma, USA
[6] University of Kentucky, Lexington, Kentucky, USA
[7] Aerospace Engineering, University of Colorado Boulder, Boulder, Colorado, USA
[8] Black Swift Technologies, Boulder, Colorado, USA
[9] Finnish Meteorological Institute, Helsinki, Finland
[10] NASA Goddard Space Flight Center, Greenbelt, Maryland, USA
[11] NOAA National Severe Storms Laboratory, Norman, Oklahoma, USA
[12] Department of Atmospheric and Oceanic Sciences, University of Colorado, Boulder, Colorado, USA
[13] National Renewable Energy Laboratory, Golden, Colorado, USA
[14] Arizona State University, Tempe, Arizona, USA
[15] Virginia Tech University, Blacksburg, Virginia, USA
[16] National Center for Atmospheric Research, Boulder, Colorado, USA

*Correspondence to*: Gijs de Boer (gijs.deboer@colorado.edu)

**Abstract.** Unmanned aircraft systems (UAS) offer innovative capabilities for providing new perspectives on the atmosphere, and therefore atmospheric scientists are rapidly expanding their use, particularly for studying the planetary boundary layer. In support of this expansion, from 14-20 July 2018 the International Society for Atmospheric Research using Remotely-piloted Aircraft (ISARRA) hosted a community flight week, dubbed the Lower Atmospheric Profiling Studies at Elevation – a Remotely-piloted Aircraft Team Experiment (LAPSE-RATE, de Boer et al., 2020a). This field campaign spanned a one-week deployment to Colorado's San Luis Valley, involving over 100 students, scientists, engineers, pilots, and outreach coordinators. These groups conducted intensive field operations using unmanned aircraft and ground-based assets to develop comprehensive datasets spanning a variety of scientific objectives, including a total of nearly 1300 research flights totaling over 250 flight hours. This article introduces this campaign and lays the groundwork for a special issue on the LAPSE-RATE project. The remainder of the special issue provides detailed overviews of the datasets collected and the platforms used to collect them. All of the datasets covered by this special issue have been uploaded to a LAPSE-RATE community set up at the Zenodo data archive (https://zenodo.org/communities/lapse-rate/).



## 1 Background

Over the past decade there has been a significant expansion of the use of unmanned aircraft systems (UAS) to make measurements of the atmosphere and its interactions with the surface of the Earth. Rapid miniaturization and cost-reductions of instrumentation and other hardware, and the aircraft platforms used to carry them, have brought UAS-based atmospheric science to many reaches of the globe. Specific advancements such as vertical take-off and landing capabilities offered by rotary wing and hybrid UAS platforms that limit the footprints required for launch and recovery (e.g. McGonigle et al., 2008; Brady et al., 2016; Hemingway et al., 2017; Wildmann et al., 2017) have made meteorological observations in focused areas feasible. Scientifically, UAS-based atmospheric science campaigns have focused on evaluating the planetary boundary layer (PBL) and atmospheric turbulence (e.g., Reuder et al., 2012; Bonin et al., 2013; Lothon et al., 2014; Altstädter et al., 2015; Lawrence and Balsley, 2013). Together, fixed- and rotary-wing UAS have expanded observations beyond those from operational meteorological observing networks (Hemmingway et al., 2017), offering high-resolution insight into key parameters in locations where surface-observations may be challenging. Specific examples of difficult-to-observe areas that have been sampled using UAS include in tropical cyclones (e.g. Cione et al., 2016; Cione et al., 2019), over the Arctic Ocean (e.g. Curry et al., 2004), in distant Antarctic regimes (e.g. Knuth et al., 2013), during a total solar eclipse (Bailey, et al. 2019), and in and around supercell thunderstorms (e.g. Elston et al., 2011). Beyond the PBL, data gathered by UAS have shed light on interactions between the atmosphere and other components of the Earth system (e.g. oceans, ice, land surface), with campaigns covering both lower latitudes (e.g. Corrigan et al., 2006; Ramanathan et al., 2007; van den Kroonenberg et al., 2008; Houston et al., 2012) and higher latitudes (e.g. Curry et al., 2004; Cassano et al., 2010; Knuth and Cassano, 2014; de Boer et al., 2018; Kral et al., 2018).

With this rise in the popularity of the use of UAS in atmospheric science, in 2008 the European Union funded a COST (Cooperation in Science and Technology) Action to advance and support the community of researchers focused on using UAS for atmospheric science. From this COST action, the International Society for Atmospheric Research using Remotely-piloted Aircraft (ISARRA) emerged. ISARRA has met annually since 2012, with five meetings in Europe (2013, 2014, 2016, 2017, 2019) and two in the United States (2015, 2018). The 2018 meeting, hosted by a joint committee of representatives from The University of Colorado, the National Center for Atmospheric Research (NCAR), and the National Oceanic and Atmospheric Administration (NOAA), was held in Boulder, Colorado, USA between 9-12 July 2018 (de Boer et al., 2019). Since 2012, several ISARRA conferences have featured coordinated flight activities to offer a field-based opportunity for participants to share information on platforms and sensing capabilities and promote collaboration between individual groups working towards similar goals. The 2018 conference offered participants a field experience highlighting Colorado-specific science topics and leveraging ongoing work within the state of Colorado to foster community relations surrounding the use of UAS. This week-long flight event, named "Lower Atmospheric Profiling Studies at Elevation – a Remotely piloted Aircraft Team Experiment (LAPSE-RATE)" was held in the San Luis Valley (SLV) of south-central Colorado immediately following the ISARRA conference (14-21 July 2018).



Prior ISARRA flight campaigns generally focused on system intercomparison and demonstrations. However, LAPSE-RATE set out to increase the coordination and scientific value of this significant effort, shifting the focus to not only offer opportunities for aircraft and sensor comparisons, but to also target data collection on specific science themes identified collaboratively by the community, including:

- *The morning boundary layer transition:* The diurnal cycle in temperature in the SLV can be substantial,
making for large swings between stable and convective boundary layers, with near-surface temperatures changing by up to nearly 25 C as part of the diurnal cycle. This topic was sampled using a distributed profiling approach, with measurement sites located geographically across different surface types and across different parts of the valley.

- *Aerosol properties:* A variety of particle sources in the SLV, including agriculture, the Great Sand Dunes
National Park, wildfires, biogenic emissions, and advection, make this an interesting area to study aerosol properties and their variability. Routine lower-atmospheric profiling allowed teams to document particle sizes and concentrations and their connections to boundary layer and synoptic wind regimes.

- *Valley drainage flows:* The SLV itself is very broad and of substantial scale (see Figure 1), with several smaller valleys feeding into it. Clear nights often result in distinct density currents from these smaller
valleys into the main SLV. Targeted sampling within the Saguache Valley and its outflow area captured the evolution of such a density current and provided data to map its outflow into the SLV.

- *Deep convection initiation:* Thunderstorms routinely form over the mountains surrounding the SLV, with storms sometimes advecting into or forming over the central part of the valley. Storm lifecycle is thought to be impacted by local sources of potential energy and/or coherent circulations tied to gradients of surface
types. Distributing teams throughout the northern portion of the valley to make detailed measurements of the thermodynamic state and its evolution with time across a variety of surface gradients offered data to evaluate whether the surface plays a role in the development and evolution of valley storms.

- *Atmospheric turbulence profiling:* Understanding atmospheric turbulence in the lower atmosphere is key to development of numerical models. Furthermore, turbulence can undermine the performance of
communication systems due to its impact on signal integrity. Teams deployed a variety of measurement platforms to characterize the diurnal cycle of turbulence intensity in the "high desert" environment of SLV to provide a unique dataset on variability within the valley, terrain effects, and the impact of low water vapor amounts on turbulence generation.

The current special issue provides information on datasets collected during the LAPSE-RATE campaign. This introductory article provides background information on the campaign itself and the general conditions that were sampled throughout the week, with additional details on the campaign and associated activities available in de Boer et al. (2020a). The subsequent articles in the special issue describe, in detail, the datasets that were collected by the various teams participating in the event. These details include information on completed flights, platforms used for
data collection, specific flight permissions, the sensors deployed and their expected accuracy, interesting conditions encountered, and an overview of statistics related to the scientific objectives and individual datasets obtained.





**2 Campaign Overview and Logistics**

The SLV is a high-altitude depositional basin located in south-central Colorado (USA) with an average elevation of 2,336 m above sea level.  The full valley stretches approximately 200 km north to south and 120 km east to west.  It is surrounded by mountains, with the Sangre de Cristo mountain range to the east and the San Juan mountain range to the west.  Both of these mountain ranges have peaks rising over 4,200 m above sea level, resulting in an altitude difference of up to 2,000 m above the valley floor.  The central valley is primarily flat, and land cover comprises

mostly agriculture and shrub/scrublands.  The agricultural areas are irrigated using groundwater and streams fed by snowmelt from the surrounding mountain ranges.  The shrub and scrubland areas include both very arid regions, as well as areas that are seasonally flooded and marshy.  Another unique feature in the valley is the approximately 600 km² Great Sand Dunes National Park, which includes a large area with expansive sand dunes, the tallest of which, Star Dune, is approximately 230 m tall from the valley floor.


To collect measurements targeting the primary objectives discussed above, teams deployed both UAS and surface-based instrumentation across the northern half of the SLV (entire area shown on map in Figure 1).  Potential sites were identified during scouting trips to the valley prior to the flight event.  The identified sites were spatially distributed and represented a variety of different land cover and terrain conditions. Flight permissions were secured from the

Federal Aviation Administration (FAA) and site access from landowners prior to the flight event. On any given day, the assembled flight teams (up to 22) from the participating organizations would set up their flight and sampling operations at one of the pre-selected sites.  While the deployment configuration changed somewhat from day to day based on the specific sampling objectives, many teams spent the majority of the campaign sampling a single location.  Additional details on which specific teams were distributed to each individual sampling location are provided in the

remainder of the articles in this special issue.  However, in the current article we provide some insight into the general motivation behind the selection of sampling locations for the different objectives covered above.

Weather conditions experienced during the LAPSE-RATE campaign were excellent for the operation of UAS and for studying the above-mentioned phenomena of interest.  Broadly speaking, the SLV was under the influence of a

developing North American Monsoon flow, whereby moisture from the Gulf of Mexico and Pacific Ocean were periodically injected into a general region of anti-cyclonic flow that varied in strength during the week-long field experiment.  High pressure was prevalent over the region during 14 July, resulting in limited convection forming over the mountains surrounding the SLV.  A cold front passed through the region on the 15th and early on the 16th. This time period also featured upper-level advection of air from the southwest, including moisture from the Pacific Ocean.

In combination, these conditions resulted in the development of more widespread convective clouds and offered favorable conditions for evaluating convective initiation.  These storms produced only limited precipitation, but did result in lightning and significant and gusty winds.   On July 17th, the upper-level anti-cyclonic circulation re-established itself over the region, reducing convective storm activity.  This quieter flow regime persisted through the



rest of the campaign, resulting in generally dry conditions, with only a limited number of thunderstorms occurring in the mountains surrounding the SLV through the end of the experiment.

Each day of LAPSE-RATE featured well-defined diurnal cycles in temperature, humidity and winds. Mornings were characterized by calm winds and elevated humidities with temperatures between 10-15 C. Localized areas of fog were observed early in the morning, particularly along the western portion of the valley. Winds were generally stronger
and gustier in the afternoon after development of the convective boundary layer, with sustained wind speeds reaching around 10 m s$^{-1}$ at times. Higher gusts associated with thunderstorm outflows were also observed locally. The weak winds that occurred overnight into late morning resulted in excellent conditions for evaluating atmospheric variability forced by local phenomena, including assessment of how surface property variability impacts boundary layer evolution and the evolution of density currents resulting from differential cooling across the valley. The spatio-
temporal variability in meteorological conditions throughout the field experiment is evident in Figure 2.

LAPSE-RATE flights began on 14 July. This first day involved an intercomparison of platforms and sensors performed at the Leach Airport, near the center of our operations area (point "E" in Figure 1). This site was chosen in part because the site features open space without nearby structures, vegetation or topography, allowing multiple
UAS teams to safely conduct flight operations at the same time. Additionally, the surrounding surface was relatively homogeneous, with the site being largely surrounded by irrigated agricultural lands and little in the way of nearby topography. Finally, there was easy access for the University of Colorado's Mobile UAS Research Collaboratory (MURC) vehicle, a platform that includes a suite of meteorological instrumentation mounted on a 15 m tower (see de Boer et al., 2020b in this special issue) that were used for the intercomparison. Teams were assigned specific flight
times in proximity to the MURC instrumentation throughout the course of 14 July. Results of this intercomparison can be found in Barbieri et al. (2019) and largely showed good agreement between the various platforms. Additional intercomparison work was done using measurements obtained by the University of Oklahoma's Collaborative Lower Atmospheric Mobile Profiling System (CLAMPS) and Doppler Lidar systems deployed by the University of Colorado (Bell et al., 2020a; Bell et al., 2020b, in this special issue); one of the University of Colorado lidars is pictured in the
foreground of Figure 1 "B".

For the remainder of LAPSE-RATE, participating teams were distributed based on the specific science objective that was deemed most attractive based on daily forecasts. This was largely decided the day before during an all-teams briefing in which weather forecasts and platform readiness was discussed. Forecast guidance for planning came
through joint input from the National Weather Service and high-resolution simulations from the Weather Research and Forecasting (WRF) model (Pinto et al., 2020; Nolan et al., 2018). Generally, days were divided into two basic categories: days where convection was expected, and days where it was not. Below, we provide overviews of the sampling completed on each day of the campaign.



• On July 15 and 16, teams were deployed to collect data to understand the development of convection over the valley. For the convective initiation and development days, teams were instructed to fly from sites covering a variety of surface types. The general idea was for all teams to conduct nearly continuous profiling at these sites to capture the thermodynamic evolution of the boundary layer and provide information on winds and turbulence (where available) resulting in the organization of convection. In addition to the teams
deploying UAS, there were the previously-mentioned surface-based remote-sensing instruments (CLAMPS, Doppler Lidars) deployed to Moffat school, Saguache Airport, Leach Airport (see Bell et al., 2020b in this special issue) as well as a variety of mobile systems that were deployed on transects throughout the valley (de Boer et al., 2020b in this special issue).

    • On July 17, teams were offered time to complete flight activities matching their own primary objectives,
without coordinated organization by the larger group. Some teams conducted additional intercomparison flights, while others conducted equipment evaluation flights to improve their sensing capabilities and expand their operational envelopes.

    • Sampling on July 18 focused on the morning boundary layer transition. During this day, teams aimed to arrive at their distributed sampling sites at sunrise (around 6:00 local time) to initiate flights and capture the
transition between the nighttime stable regime and the development of a surface-radiation driven convective boundary layer, with sampling continuing until around mid-day. Teams were once again distributed across the valley at sites featuring a variety of surface types, with some teams operating on eastward slanting slopes, and others on westward-slanting slopes. Surface-based assets were again located at Moffat School, Leach Airport and Saguache Airport.

• Finally, July 19 focused on cold-air drainage flows that set up during the night-time. For this effort, teams were distributed with a focus on the Saguache Valley that feeds into the northwest corner of the SLV (around point "B" in Figure 1). Several teams were distributed in the Saguache Valley itself, including at the Saguache Airport which has an Automated Weather Observing System (AWOS). An additional array of UAS platforms was deployed in the block of agricultural fields situated at the outflow region (near point "C" in
Figure 1). Further, some teams were situated within different parts of the main valley, and two teams were assigned to conduct flights in the valley feeding into the northern portion of the SLV (near point "A" in Figure 1).

    • July 20, the final day of the LAPSE-RATE campaign, was once again open for teams to conduct their own sampling activities throughout the SLV as needed to accomplish individual goals and tasks. Some teams
conducted additional comparison flights with the MURC on this date.

Finally, given that only a limited number of teams were interested in studying aerosol processes during the campaign, these measurements were made continuously at a single site approximately 15 km north of Leach Airport, both using UAS-based sensors as well as from the surface (see Brus et al., 2020 in this special issue).




### 3 Overview of Datasets

Participants in the LAPSE-RATE campaign spanned a variety of organizations including various U.S. and international universities, U.S. federal agencies, U.S. and international institutes, and various companies in the private sector. An overview of the groups responsible for the collection of data during LAPSE-RATE, along with their specific roles and the special issue publications that are connected to those activities is provided in Table 1. As mentioned above, these groups include operators of unmanned aircraft, surface-based in situ and remote sensor systems, radiosondes, and surface vehicles. Additionally, they include modelers, forecasters, and outreach teams.

All of the datasets have been quality-controlled based on protocols established by the individual teams. The details behind this quality control are outlined in the individual articles and will not be explained in detail here. However, our common goal was to provide a quality dataset that could be leveraged by the community for scientific studies. Data are provided in NetCDF format with a common file name structure. This structure is:

xxx.ppppp.lv.yyyymmdd.hhmmss.cdf

using the following definitions:

| | |
|---|---|
| xxx: | Institute (see appendix 1) |
| ppppp: | A 5-letter platform identifier |
| lv: | The data file processing level |
| hhmmss: | The file start time (UTC) in hours, minutes, seconds |
| yyyymmdd: | The file date (UTC) in year, month, day, month |
| cdf: | The NetCDF file extension |

As an example, a University of Colorado Boulder produced NetCDF file from the DataHawk UAS that includes non-quality-controlled data (in geophysical units) collected starting at 11:30:54 UTC on 17 July, 2018 would be named UCB.DATHK.a1.20180717.113054.cdf.

The groups provided files at various levels of processing. These include:

**a0:** Raw data converted to netCDF

**a1:** Calibration factors applied and converted to geophysical units

**b1:** QC checks applied to measurements to ensure that they are "in bounds". Missing data points or those with bad values should be set to -9999.9

**c1:** Derived or calculated value-added data product (VAP) using one or more measured or modeled data (a0 to c1) as input

Note that it was not required (or expected) that all groups would submit all data levels. For example, a0 data may be deemed too detailed, too large, and without sufficient quality control to be useful to many interested users and therefore may not be made available by many of the participating groups. All groups sought to provide a1 or b1 data products





where possible as this level of processing offers the greatest potential for community consumption and ensures some level of quality control.


Participating teams were requested to provide sufficient metadata in their NetCDF files to allow users to understand and interpret the data stream. This includes information that could be included in global metadata (i.e. specific to the entire file) such as:

- Location of sensor/platform (if stationary platform)

- QC checks and flags applied

- Calibration procedural information as appropriate

- PI Contact information

Additionally, for individual variables, it was requested that teams provide metadata to help the user understand how the measurement was obtained and what it represents. This type of information could include:

- Orientation: downwelling, upwelling, or dependent on installation as appropriate.

- Missing data value applied to a given variable (e.g. -9999.9)

- Key information to characterize the measurement

- The instrument/sensor used for the measurement (occasionally important especially if it comes from a data stream containing results from several instruments).

- Time interval information (e.g., averaging time and measurement intervals).

- The units of the measurement

- An indication that the data is a best estimate data or a calculated value data stream. Unless indicated otherwise, it is implicit that the measurement is observed.

Files from all teams are archived under individual DOIs at the Zenodo data archive (www.zenodo.org), where a dedicated community has been established for LAPSE-RATE (https://zenodo.org/communities/lapse-rate/). This community houses the data files along with additional metadata on the datasets. In total, these files cover nearly 1300 flights and 250 flight hours, along with data from related ground-based observing systems, radiosondes, and numerical model output.

**4 Summary**

During the summer of 2018, a large group of atmospheric scientists and engineers came together in the San Luis Valley of Colorado to conduct a coordinated atmospheric measurement program using unmanned aircraft. This group deployed fixed- and rotary-wing UAS, in combination with surface based in situ and remote sensors, and radiosondes to make comprehensive measurements to demonstrate the utility of these platforms in understanding a variety of

boundary-layer phenomena including the morning boundary-layer transition, density currents, convection initiation, aerosol processes, and turbulence. Additionally, intercomparison efforts were undertaken to evaluate to what extent these types of measurements can be used together. In the end, a large number (>1200) of flights were completed. This article provides an overview of the campaign, the conditions sampled, and the general structure of data collection.

This overview introduces this special issue on the LAPSE-RATE campaign, which hosts individual articles from the teams involved that provide additional details on their platforms, sensors and flights. We hope that this issue will encourage widespread use of the data and encourage individuals to reach out to the teams who collected the datasets if they have interest in using them for their own scientific purposes.

**Acknowledgments.** LAPSE-RATE was made possible by the active participation of over 100 people. Limited financial support, specifically for participant travel, was provided by the U.S. National Science Foundation (NSF) (AGS 1807199) and the U.S. Department of Energy (DE-SC0018985). Partial support was also provided by the U.S. National Science Foundation through award #1539070, Collaboration Leading Operational UAS Development for Meteorology and Atmospheric Physics (CLOUDMAP) and through award no. CBET-1351411. Further, this work

was authored in part by the National Renewable Energy Laboratory, operated by Alliance for Sustainable Energy, LLC, for the U.S. Department of Energy (DOE) under Contract No. DE-AC36-08GO28308. The views expressed in the article do not necessarily represent the views of the DOE or the U.S. Government. The U.S. Government retains and the publisher, by accepting the article for publication, acknowledges that the U.S. Government retains a nonexclusive, paid-up, irrevocable, worldwide license to publish or reproduce the published form of this work, or

allow others to do so, for U.S. Government purposes. We appreciate and acknowledge the long hours dedicated to this effort by numerous participants not included in the author list of this overview document, including many undergraduate and graduate students and other early-career participants. We also would like to thank the San Luis Valley community for welcoming us and providing access to land to conduct flight operations, and recognize the sponsors indicated in the Acknowledgement sections of the remaining special issue articles for their support of this

effort.

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



**Table 1: A list of data-generating entities participating in LAPSE-RATE and the individual roles that each entity filled during the campaign.**

| Organization | Primary Role | Relevant Publications [ * In this ESSD special issue ] |
|---|---|---|
| University of Colorado Boulder | Campaign coordination, operation of unmanned aircraft, deployment of surface-based Doppler lidar systems and mobile measurement system | de Boer et al., 2020b* de Boer et al., 2020c* Bell et al., 2020b* |
| University of Oklahoma | Operation of unmanned aircraft, deployment of CLAMPS surface observatory | Pillar-Little et al., 2020* |
| University of Kentucky | Operation of unmanned aircraft, deployment of surface flux tower | Bailey et al., 2020* |
| Oklahoma State University | Operation of unmanned aircraft | Natalie et al., 2020* |
| University of Nebraska – Lincoln | Operation of unmanned aircraft, deployment of mobile measurement systems | Islam et al., 2020* |
| Virginia Tech | Operation of unmanned aircraft | TBD* |
| Kansas State University | Operation of unmanned aircraft, deployment of surface aerosol instrumentation | Brus et al., 2020* |
| National Oceanic and Atmospheric Administration (Physical Sciences Laboratory, National Severe Storms Laboratory, Chemical Sciences Laboratory) | Campaign coordination, campaign forecasting, launching of weather balloons, deployment of mobile measurement system | de Boer et al., 2020c* Bell et al., 2020b* |
| National Center for Atmospheric Research | Campaign forecasting, production of numerical forecast products, modeling and post-campaign support | Pinto et al., 2020* |
| Finnish Meteorological Institute | Operation of unmanned aircraft | Brus et al., 2020* |
| Black Swift Technologies | Operation of unmanned aircraft | de Boer et al., 2020b* |

430

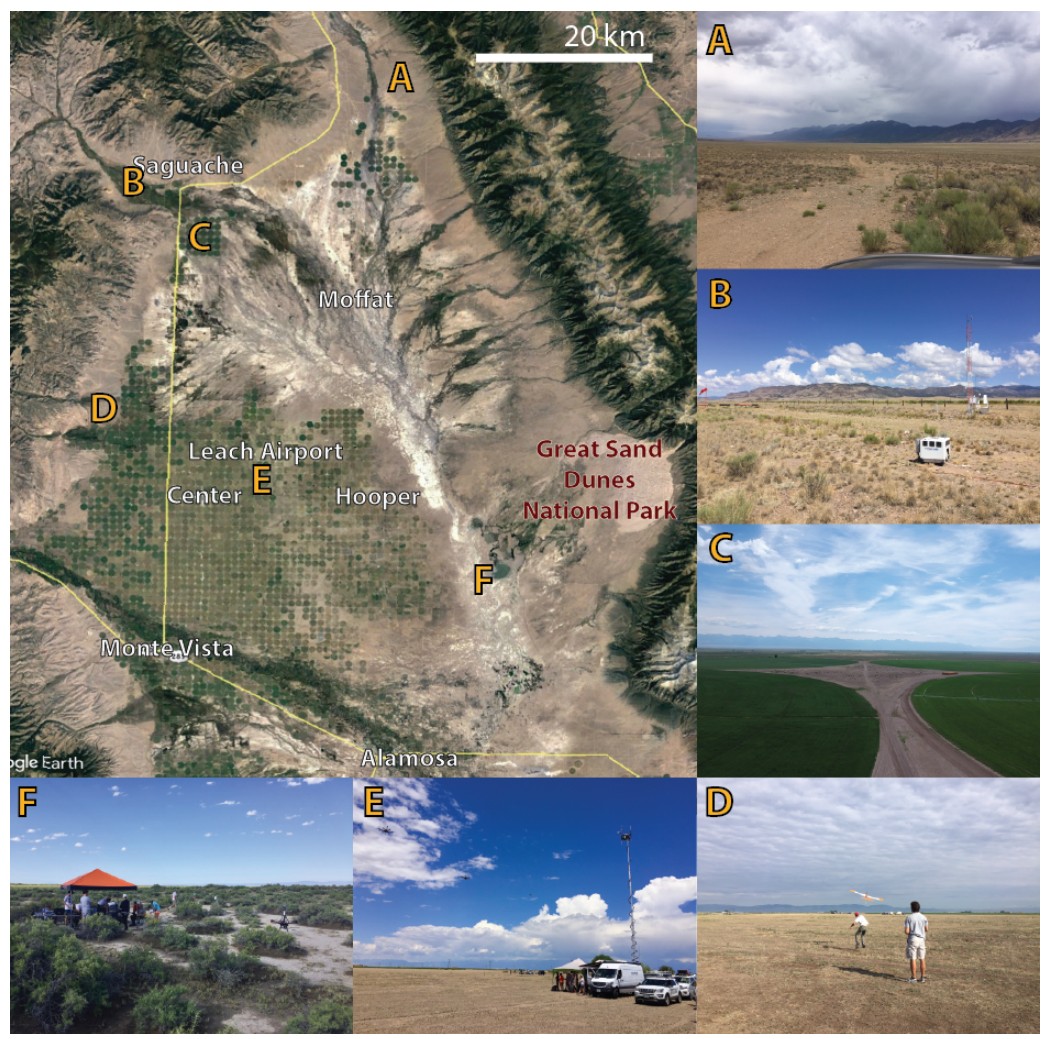

**Figure 1:** A map of the San Luis Valley of Colorado, and photographs providing a perspective on the general variability of
435 surface conditions around the valley. The satellite image in Figure 1A is courtesy of Google Maps.

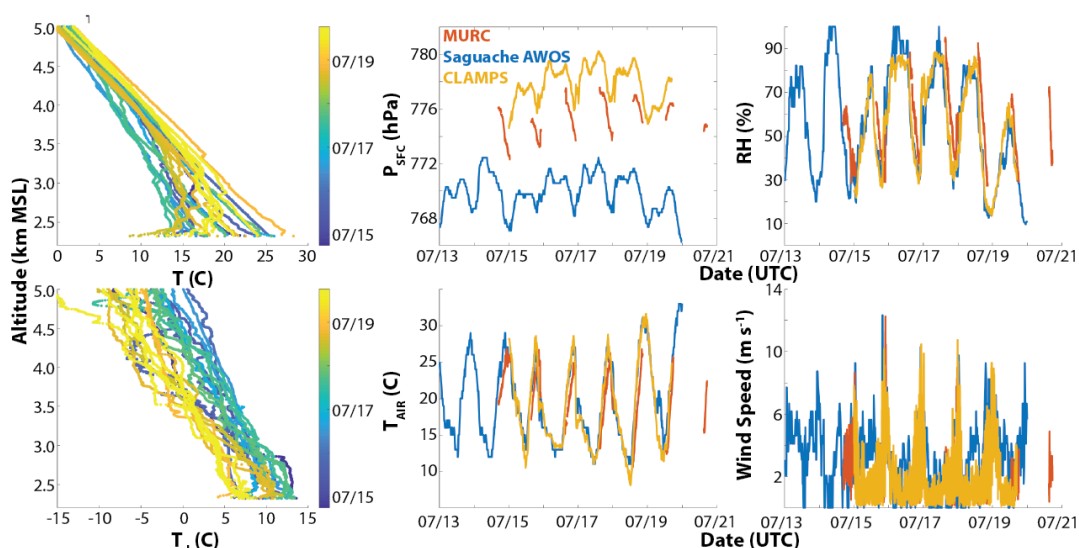

**Figure 2:** An overview of meteorological conditions observed in the San Luis Valley during LAPSE-RATE. The two leftmost figures illustrate temperature (top left) and dewpoint (bottom left) profiles obtained from radiosondes launched at Leach Airport, with the colors representing the date of launch. The center and righthand panels illustrate surface meteorology at three different sites across the valley over the course of the week. These include surface pressure (top center), air temperature (bottom center), relative humidity (top right) and wind speed (bottom right). These observations were obtained by the MURC vehicle (red), the Saguache Airport AWOS station (blue) and the CLAMPS trailer (yellow).

440