# Peer review of "Data Generated During the 2018 LAPSE-RATE Campaign: An Introduction and Overview"

_Earth System Science Data, 2020_

## Referee Comment (RC1) · Anonymous Referee #1 · 10 Jul 2020

This paper overviews the LAPSE-RATE field campaign and the datasets obtained by the campaign in details. The dataset is complete and it is very high quality. The manuscript is written clearly. So this paper may be published without any revision except for the following typo.

L245: yyyymmdd: year, month, day, month –> No need for last 'month'

L244-245: It is better to reverse the order of hhmmss and yyyymmdd.
* * *

---

## Referee Comment (RC2) · Anonymous Referee #2 · 23 Jul 2020

Very good product, very well written. Overview serves the constituent data sets, available from Zenodo. Good data organization and - one presumes without yet seeing many of the individual data sets - good formats and metadata. Zenodo links work well.

Three small comments:

1) In line 79, authors confuse readers by referring to dates of the flight/measurement campaign (14-21 July 2018) when - technically - their text refers to the ISARRA conference. E.g. conference happened before, measurement period happened after (14-21 July). Dates of data gathering become clear later in the manuscript but authors could revise this sentence slightly to remove ambiguity.

2) Lines 313 to 316, presumably the standard US govt license applies only to US-

funded investigators? Or to the entire data set? E.g. has FMI agreed to this license?

3) In legend for Figure 1 (e.g. line 436), authors write "The satellite image in Figure 1A is courtesy of Google Maps". But Figure 1A shows an (eastward?) view from the north end of SLV. The satellite image - quite obvious in this collage - does not correspond to 1A? Authors could simply say "The satellite image is courtesy of Google Maps" because they only show one quite-obvious satellite image.

---

## Author Comment (AC1) · 27 Aug 2020

**Response to reviewer comments for ESSD-2020-98, "**Data Generated During the 2018 LAPSE-RATE Campaign: An Introduction and Overview"

Below, we address the comments provided by the reviewers for this manuscript. We would like to thank both of the reviewers for their time and for their feedback. Reviewer comments are in black, and responses below are in red.

**Reviewer #1:**

This paper overviews the LAPSE-RATE field campaign and the datasets obtained by the campaign in details. The dataset is complete and it is very high quality. The manuscript is written clearly. So this paper may be published without any revision except for the following typo.

We would like to thank the reviewer for the time spent reading through the manuscript, and for their constructive feedback.

L245: yyyymmdd: year, month, day, month –> No need for last 'month'

Thank you for pointing this out. It was a typo, and the second "month" has been removed.

L244-245: It is better to reverse the order of hhmmss and yyyymmdd.

Thank you again – this makes a lot of sense. We have reversed the order of these two lines.

**Reviewer #2:**

Very good product, very well written. Overview serves the constituent data sets, available from Zenodo. Good data organization and - one presumes without yet seeing many of the individual data sets - good formats and metadata. Zenodo links work well.

We would like to thank the reviewer for the time spent reading this manuscript and for the kind comments.

Three small comments:

1) In line 79, authors confuse readers by referring to dates of the flight/measurement campaign (14-21 July 2018) when - technically - their text refers to the ISARRA conference. E.g. conference happened before, measurement period happened after (14-21 July). Dates of data gathering become clear later in the manuscript but authors could revise this sentence slightly to remove ambiguity.

Thank you for this comment. We have reworded that sentence to say:

*This week-long flight event, named "Lower Atmospheric Profiling Studies at Elevation – a Remotely piloted Aircraft Team Experiment (LAPSE-RATE)" was held in the San Luis Valley (SLV) of south-central Colorado from 14-21 July, 2018, immediately following the 2018 ISARRA conference.*

We believe that this should take care of the reviewer's concerns about the dates being confusing.

2) Lines 313 to 316, presumably the standard US govt license applies only to US-funded investigators? Or to the entire data set? E.g. has FMI agreed to this license?

These lines were added in response to a request from the National Renewable Energy Laboratory (NREL), part of the US Department of Energy. The government license is specific to the publication, not to the datasets themselves, which have their own DOIs.

3) In legend for Figure 1 (e.g. line 436), authors write "The satellite image in Figure 1A is courtesy of Google Maps". But Figure 1A shows an (eastward?) view from the north end of SLV. The satellite image - quite obvious in this collage - does not correspond to 1A? Authors could simply say "The satellite image is courtesy of Google Maps" because they only show one quite-obvious satellite image.

Yes, this is confusing – we apologize for this oversight. In fact, there is no "Figure 1A". The photograph labeled with "A" is indicating what site "A" on the map looked like. We have reworded the last sentence of the caption to clarify where the Google Maps satellite imagery is as:

*The satellite image in the large map in the upper left is courtesy of © Google Maps.*